# The estimated benefits of increasing cigarette prices through taxation on the burden of disease and economic burden of smoking in Nigeria: A modeling study

Ariel Bardach[1,2]*, Agustín Casarini[1], Federico Rodriguez Cairoli[1], Adedeji Adeniran[3], Marco Castradori[3], Precious Akanonu[3], Chukwuka Onyekwena[3], Natalia Espinola[1], Andrés Pichon-Riviere[1,2], Alfredo Palacios[1]

**1** Institute for Clinical Effectiveness and Health Policy (IECS-CONICET), Buenos Aires, Argentina, **2** Center for Research in Epidemiology and Public Health, National Scientific and Technical Research Council (CONICET), Buenos Aires, Argentina, **3** Center for the Study of the Economies of Africa, Abuja, Nigeria

* abardach@iecs.org.ar

## Abstract

### Background

Globally, tobacco consumption continues to cause a considerable burden of preventable diseases. Although the smoking prevalence in Nigeria may be declining over the last years, the absolute number of active smokers remains one of the highest in Africa. Little is known about the disease burden and economic costs of cigarette smoking in Nigeria. Consequently, there is an evidence gap to inform the design and implementation of an effective policy for tobacco control.

### Methods

We applied a microsimulation model to estimate the burden attributable to smoking in terms of morbidity, mortality, disability-adjusted life-years (DALYs), and direct medical costs and indirect costs (e.g., productivity loss costs, informal caregivers' costs). We also modeled the health and economic impact of different scenarios of tobacco price increases through taxes.

### Results

We estimated that smoking is responsible for approximately 29,000 annual deaths in Nigeria. This burden corresponds to 816,230 DALYs per year. In 2019, the total economic burden attributable to tobacco was estimated at ₦ 634 billion annually (approximately U$D 2.07 billion). If tobacco cigarettes' prices were to be raised by 50% through taxes, more than 30,000 deaths from smoking-attributable diseases would be averted in 10 years, with subsequent savings on direct and indirect costs of ₦597 billion and increased tax revenue collection of ₦369 billion.

**Data Availability Statement:** All relevant data are within the paper and its Supporting Information files.

**Funding:** Yes. Funded by International Development Research Centre (IDRC) Project Number: 108825 Organization who received the award: Centre for the study of the economies of Africa (CSEA) The funders had no role in study design, data collection and analysis, decision to publish, or preparation of the manuscript.

**Competing interests:** NO authors have competing interests

## Conclusion

In Nigeria, tobacco is responsible for substantial health and economic burden. Increasing tobacco taxes could reduce this burden and produce net economic benefits.

## Introduction

In 2019, 7.7 million deaths and 200 million disability-adjusted life-years (DALYs) were attributed globally to tobacco [1]. Nigeria, the most populous country in Africa, is currently leading the tobacco market in Africa, with more than 18 billion cigarettes sold annually [2]. The American Cancer Society's Tobacco Atlas estimated that more than seven million adults are daily smokers in Nigeria for 2015, with more than 300 deaths per week attributable to smoking [3]. Despite recent national initiatives targeted at reducing and regulating the use of tobacco products in the country (e.g., the National Tobacco Control Act of 2015), which in turn reinforces the prerogatives of the World Health Organization Framework Convention on Tobacco Control (WHO FCTC) in 2006 [4], the absolute number of active smokers remains one of the highest in Africa [2]. A recent meta-analysis of 64 studies by Adeloye et al. reports that the pooled prevalence of current smokers in Nigeria was 10.4% (9.0–11.7), which is only 3% under the regional prevalence [5], and that of ever smoking was 17.7% (15.2–20.2) [2].

Because of the large population size and access to other markets in the region, Nigeria is a key tobacco industry market in Africa. The British American Tobacco (BAT) has been trading in Nigeria since 1911, with its operations intensifying after establishing the Nigerian Tobacco Company (NTC) in 1951—a manufacturing, distributing, and marketing company jointly owned by the Nigerian Government and BAT. As recently as 2003, with great encouragement from the Federal Government, BAT built a US$150 million state-of-the-art (implying lower employment needs) manufacturing plant in Nigeria to service West African countries and opened its new West Africa Head Office in Lagos in 2016 [6]. While Nigeria's market size justifies its attractiveness as a destination for tobacco multinationals, Nigeria's history of weak development of anti-tobacco laws and initiatives has undoubtedly contributed. Relatively loose regulations and uncertain enforcement characterize Nigeria's tobacco control policy, creating a favorable environment for licit and illicit products traders.

In the country, decision-makers lack information on the burden of disease and economic burden attributable to tobacco consumption, such as the annual health events and deaths of tobacco-related conditions, direct medical costs, and indirect costs (borne by patients and society). Decision-makers also need other sensitive information to implement tobacco control interventions, such as the effectiveness of tobacco tax policies and other tobacco control measures, and the benefits obtained from them (deaths and direct and indirect costs avoided, fiscal revenues in the case of tobacco tax, etc.).

This study's objective is to estimate the tobacco-related burden of disease, its direct and indirect costs, and evaluate the health and financial impact of different cigarette price levels increase through taxes in Nigeria.

## Methods

The Institute for Clinical Effectiveness and Health Policy (IECS) coordinated a multi-country initiative to develop an economic model to estimate the tobacco-related disease and economic burden and evaluate the impact of different tobacco control interventions, including taxation,

cigarette plain packaging, advertising, and smoke-free environments [7]. This model has been applied in several studies to estimate the burden of smoking and the potential impact of tobacco control interventions in different countries [8–13].

The model corresponds to a first-order Monte Carlo simulation, which follows a hypothetical cohort throughout its lifetime [7]. The model estimates various outcomes such as disease incidence, quality of life, disease events, and healthcare costs for each sex and age strata for smokers, ex-smokers, and never smokers. By incorporating the natural history, costs, and quality of life of all the tobacco-related adult-specific diseases, the model allows for a mock-up of individuals' lifetimes in hypothetical cohorts. Health outcomes will occur according to annual risk equations based on their smoking status. The risk of acute and chronic events is estimated from the baseline risk in non-smokers multiplied by the age, gender, and condition-specific relative risks (RR) for smokers and ex-smokers [14].

The risk of death was defined according to the events, and conditions individuals suffered, including general mortality by sex and age. Finally, using previously determined parameters of quality of life and healthcare costs, we estimated the quality-adjusted life-years (QALYs) and total costs for the cohort's overall survival time, respectively. The study used the DALY approach to decompose years of life lost due to premature mortality (YLL) and years lost due to disability (YLD). However, DALYs were not age-weighted, and no discount was applied. To estimate YLD, we used utility values identified through extensive literature searching, where disability weights are equal to 1 –utility, while YLL was derived from life tables. The health conditions considered were coronary (ICD-10 code: I20; I21-22; I24-25) and non-coronary heart disease (I00;I010-I012;I018-I020;I029;I050-I052;I058;I062;I068-I072;I078-I083; I088-I092;I098-I099;I110;I119;I260;I269-I272;I278-I281;I288;I289;I300;I301;I308-I313; I318-I319;i320;I321;I328;I330;I339-I342;I348-I352;I358-I362;I368-I372;I378;I379;I38X; I390-I394;I398;I400;I401;I408;I409;I410-I412;I418;I420-I429;I430-I432;I438;I440-I447; I450-I456;I458-I461;I469-I472;I479;I48X;I490-I495;I498-I501;I509-I519;I059.I060-1; I700-I702; I708;I709), cerebrovascular disease(I600-I629;I630-I639;I64;I678;I679;I690-I694; I698); chronic obstructive pulmonary disease—COPD(J40-J43)—; pneumonia (J10-J18); leukemia (C92.0), lung (C34), mouth and pharynx(C000-C009;C140;C142;C148), larynx(C32), esophagus(C150-C159), stomach(C160-C169), pancreas(C250-C259), kidney(C64), bladder (C67), and cervix cancer(C53).

Although the model does not assess the consequences of passive smoking and the main smoking-related perinatal causes (low weight or low size at birth, respiratory distress syndrome, and sudden infant death syndrome) directly, the potential years of life lost, deaths, and costs associated with it were incorporated using estimates reported in the US studies [15]. Hence, an additional burden of 13.6% in men and 12% in women over direct estimations was applied, based on studies of the U.S. Department of Health and Human Services [16].

We analyzed differences in the total absolute numbers and rates of events, deaths, and associated costs to quantify the smoking-attributable disease and economic burdens, considering current Nigeria (with the current prevalence of smokers and ex-smokers) minus a 'hypothetical Nigeria' in which tobacco smokers never existed.

The IECS model also allows the simulation of the effect of different strategies aimed at tobacco control, such as increasing cigarette taxes. We explored three scenarios of tobacco price increases through taxes, corresponding to 25%, 50%, and 75% total price increases over a time spam of 10 years. Thus, changes in the prices would reduce the tobacco consumption trough the price elasticity of demand, and finally the change in consumption would impact on the tobacco prevalence as it is shown in the following formula. Furthermore, the model allows an adjustment by possible illicit trade effects. The effect of these price increases on the

prevalence of smoking was calculated as:

$$Prevalence = Prev_B + (\alpha * \varepsilon_d + (1 - \alpha)\varepsilon_{cp}) * \Delta P * I_p * Prev_B$$

Where $Prev_B$ is the baseline prevalence of smoking before price increase; $\alpha$ is the market share of licit tobacco products; $\varepsilon_d$ is the price elasticity of demand for tobacco products; $\varepsilon_{cp}$ is a pseudo cross-price elasticity of demand between illicit and legal cigarettes (obtained from literature [17]); $\Delta P$ is the percentage change in price for each scenario (25%, 50% or 75%); and $I_p$ is the proportion of the variation on cigarette consumption expected to impact on smoking prevalence, that in the short term, the first 5 years of the simulation, it was assumed that 50% of the reduced consumption is a consequence of the reduction in prevalence ($I_p$. = 0.5) to represent as conservative scenario, while in the long run the $I_p$ would be assume equal to 75% representing a greater impact of the price change over the prevalence. More details are presented elsewhere [13].

Finally, the percentual effect over the tax revenue ($\Delta\%R$) was estimated as the multiplication of the change in the consumption times proportion of the price increase that correspond to taxes, measured by the coefficient $\frac{\Delta P}{\%P_{tax}}$, where $\%P_{tax}$ represent the percentage of the price that are taxes.

$$\Delta\%R = (1 + \varepsilon_d * \Delta P) * \left(1 + \frac{\Delta P}{\%P_{tax}}\right) - 1$$

## Epidemiological methods and data

Regarding epidemiological data, local sources of good quality were the first choice; international sources were used as a second option when these were not available. The probability of acute events, the incidence of chronic diseases and their progression, and mortality rates associated with the conditions analyzed by age and sex, were drawn mainly by coupling estimations from local and international sources. On the one hand, local data on costs of managing the different conditions were obtained from three public referral hospitals in Nigeria (National Hospital Abuja (NHA), University College Hospital (UCH), Ibadan, and University of Nigeria Teaching Hospital (UNTH), Enugu State). On the other hand, the International Agency for Research on Cancer (IARC) Cancer Today database [18] and the Institute of Health Metrics' (IHME) Global Burden of Disease project (GBD) [19] were the international sources utilized for cancer incidence and specific mortality from related conditions, respectively.

For this model, Nigerian demographic data for the population over 35 years of age was considered [20]. Data on the prevalence of smoking and ex-smoking was introduced in the model for the target population [21]. For each condition included in the model, we used data regarding the incidence, prevalence, case fatality rate, and the total number of deaths [19]. Epidemiological parameters were calibrated for cancer diseases considering country-specific data on diagnosis and survival [18]. Likewise, the most representative relative risk value was used for each of the conditions regarding the subgroup of smokers, former smokers, and non-smokers [22] (see **S1 Appendix**). Finally, several international sources reporting utility values on a 0–1 scale for the construction of QALYs were also used [23–37] (see **S2 Appendix**). Regarding economic parameters, the own-price elasticity (-0.496) [38],the cross-price elasticity between licit and illicit tobacco (0.17) [17] and tobacco tax revenue in local currency, which is Nigerian Naira (₦), (₦36,3 billion) were obtained from previous studies [39]. Further economic parameters needed for comparison purposes were extracted from the World Development Indicators [40] considering the latest available data at July 2020: Nigerian GDP (₦145,639 billion), National health expenditure as a percentage of GDP (3.76%), and exchange rate (1 U$D = ₦306.92).

**Table 1** summarizes information about the total population and percentage of current/former smokers by gender and age groups (for the detailed prevalence of current/former smokers and the entire population by single ages and gender see **S3 Appendix**).

**Table 1. Total population and smoking prevalence by gender and age groups in Nigeria (GATS 2012, Nigeria).**

| Age group | Men | | | Women | | |
| --- | --- | --- | --- | --- | --- | --- |
| | Total Population (number) | Current Smokers (Prevalence) | Ex-smokers (Prevalence) | Total Population (number) | Current Smokers (Prevalence) | Ex-smokers (Prevalence) |
| 35–44 | 9,257,215 | 8% | 5% | 9,730,940 | 5% | 2% |
| 44–65 | 10,298,790 | 11% | 9% | 8,095,575 | 5% | 3% |
| > = 65 | 3,490,399 | 8% | 20% | 2,757,322 | 9% | 12% |

## Direct and indirect costs methods and data

The direct medical cost of events attributable to tobacco consumption was estimated using two complementary methodologies based on the availability of local data. First, a micro-costing approach was used for the estimation of the costs on the first year of the following conditions: coronary and non-coronary heart disease; cerebrovascular disease; moderate chronic obstructive pulmonary disease (COPD); pneumonia; lung, mouth, larynx, pharynx, esophagus, stomach, pancreas, kidney, bladder, and cervix cancer; and leukemia. Second, the costs of mild and severe COPD, those of stroke follow-up, long-term follow-up for cancer-related costs, were estimated using an indirect approach based on the extrapolation from previous research done in Latin American countries [13] with socioeconomic characteristic like those of Nigeria, as population, GDP per capita and health expenditure.

For micro-costed events, we considered the estimations made by the Center of Studies of the Economies of Africa (CSEA), where the data was primarily collected from four hospitals over three Nigerian regions with the purpose of covering three distinct geopolitical and cultural zones across the country, namely: Oyo (Southwest), Enugu (Southeast), and Abuja (North). Based on access to treatment, these institutions are the main facilities in their respective region and people seeking care adequately reflect the vast social and economic differences that exist throughout the country. The procedure employed for primary cost collection consisted of two steps. First, interviews with physicians and experts on smoking-related diseases were carried out to obtain the list of healthcare resources used, including medical, pharmacological, lab exams, etc. Then, each resource's price was gathered from health centers or pharmacies according to each resource. Finally, to provide results at the national level, the event cost of each hospital was weighted considering the population size of each region. The direct medical costs for the conditions considered are shown in **Table 2**.

The model also considered the indirect costs attributable to tobacco consumption: the productivity loss costs and informal caregivers' costs. For the former, we computed the productivity losses by considering two factors. Firstly, due to premature death costs, which add up to the wages, a person cannot earn during their working life due to death caused by a tobacco-attributable disease. Secondly, productivity losses due to disability are considered that individuals' work productivity decreased due to smoking at the same proportion as the reduction of quality of life attributed to it [41]. The pricing of these losses was calculated according to the actuarial formula of the value of a statistical life [10]:

$$VSL = \sum_{j=i}^{E(x)} prob(alive) * wage * \left(\frac{1+g}{1+r}\right)^{E(x)-j}$$

In which *prob(alive)* is the probability that an individual will be alive the following year; *wages* is an estimate of the individual's annual income from work, that in the case of Nigeria was estimated using household expenditure data from the General Household Survey Panel [42], considering that any database contains information on household income by age and

**Table 2. Estimated direct medical costs (in ₦ as of March 2020).**

| Disease events (annual) | Cost (₦) | Method/source |
|---|---|---|
| Acute myocardial infarction (AMI) | 402.411 | Microcosting |
| Non-AMI ischemic event | 1.173.994 | Microcosting |
| Stroke | 1.208.400 | Microcosting |
| Pneumonia/influenza | 61.249 | Microcosting |
| Moderate COPD (annual) | 232.556 | Microcosting |
| Lung cancer 1st year | 3.851.526 | Microcosting |
| Mouth cancer 1st year | 1.714.859 | Microcosting |
| Esophageal cancer 1st year | 1.264.945 | Microcosting |
| Stomach cancer 1st year | 1.266.866 | Microcosting |
| Pancreatic cancer 1st year | 1.918.056 | Microcosting |
| Kidney cancer 1st year | 1.525.267 | Microcosting |
| Laryngeal cancer 1st year | 1.792.030 | Microcosting |
| Leukemia 1st year | 2.650.265 | Microcosting |
| Bladder cancer 1st year | 1.241.534 | Microcosting |
| Cervical cancer 1st year | 2.446.750 | Microcosting |
| CHD follow-up (annual) | 193.106 | Indirect estimations |
| Stroke follow-up (annual) | 363.348 | Indirect estimations |
| Mild COPD (annual) | 86.782 | Indirect estimations |
| Severe COPD (annual) | 3.863.457 | Indirect estimations |
| Lung cancer 2nd year | 4.709.201 | Indirect estimations |
| Mouth cancer - 2nd year onwards | 1.277.677 | Indirect estimations |
| Esophageal cancer - 2nd year onwards | 936.001 | Indirect estimations |
| Stomach cancer - 2nd year onwards | 1.072.434 | Indirect estimations |
| Pancreatic cancer - 2nd year onwards | 1.540.811 | Indirect estimations |
| Kidney cancer - 2nd year onwards | 1.074.893 | Indirect estimations |
| Laryngeal cancer - 2nd year onwards | 705.926 | Indirect estimations |
| Leukemia - 2nd year onwards | 3.153.983 | Indirect estimations |
| Bladder cancer - 2nd year onwards | 977.246 | Indirect estimations |
| Cervical cancer - 2nd year onwards | 1.828.462 | Indirect estimations |

COPD: chronic obstructive pulmonary disease, CHD: coronary heart disease

* Exchange rate per dollar 1 U$D = 306.92 NGN.

gender, the salary was computed as the annual household expenditure per worker. The last term considers two parameters assumed as constants: a growth rate over time in income from work (parameter $g$), the premise of which is that this growth is equal to the mean annual growth rate for Nigeria's per capita GDP, or 1.21% per annum, from 1960 to 2019 [40], this parameter captures the trend of economic growth of Nigeria, and a 5% discount factor for future income (parameter $r$). Calculation of the VSL associated with an individual of a given sex and age is the sum of the products for each age until the retirement age (according to Nigerian civil service decree No. 43 of 1988 is 60 years for men and women).

Regarding the latter, we estimated the total hours of informal care needed for each health event through a literature review [43–52] and for the cases in which data were not obtained from the literature review, an econometric estimation was performed to estimate the missing data indirectly. The model was based on the relationship between the utility associated with the diseases included in the model and the hours of informal care per day per illness, identifying that a disease with less utility corresponds to a more significant number of hours of

informal care. Also, information was validated with formal caregivers. Then, we valuated these hours using the opportunity cost approach [53], considering the average expenditure of workers as a proxy of the cost of the informal caregiver [42]. Previous studies held in Nigeria have reported that informal caregivers are usually married women who take care of their partner, of whom usually have reached a secondary educational level, and they have concluded that informal caregivers suffer not only financial burdens and strains but also social, emotional, health aftermaths [54–56].

## Results

### Deaths and events

Our model estimated that approximately 29,000 deaths are attributable to smoking in Nigeria annually, representing around 16% of total deaths from smoking-related diseases in the country (183,000).

COPD was the leading cause of smoking-related mortality (29%) followed by ischemic heart disease (17.5%), stroke (13%), passive smoking (11.5%), lower respiratory tract infection (11%), and cardiovascular deaths of non-ischemic origin (5.5%). In aggregated terms, COPD (29%) was the most prevalent disease group, followed by cardiovascular disease (23%).

For the conditions analyzed, nearly 737,000 events are expected to occur every year, of which 128,000 (17%) would be attributable to cigarette consumption. COPD is the condition with the higher figure of attributable events 68,937 (54%) followed by pneumonia with 31,663 (24%) and stroke and cardiovascular diseases with almost 11,150 (9%) each. We show the main results of the burden of disease attributable to cigarette consumption in **Table 3.**

### DALYs (premature mortality and disability)

In Nigeria, smoking causes 816,230 DALYs. Of this total, 77% is caused by premature mortality, and the remainder is caused by disability. Men account for 69% of the DALY burden. Based on a simulated cohort of 35 years of age with Nigerian life expectancy, Table 4 shows the mean differential QALYs by gender for never-smokers and smokers, as well as the mean overall DALYs for smokers and ex-smokers. Tobacco-related deaths were primarily caused by COPD (29%) followed by ischemic heart disease (17.5%), stroke (13%), passive smoking (11%), lower respiratory tract infection (11%) and non-ischemic cardiovascular deaths (5.5%). Among all disease groups, COPD (29%) and cardiovascular disease (23%) ranked first and second, respectively. If, in addition, passive smoking and other causes not currently included in the model, like perinatal disease and accidents related to smoking, were considered, the value would rise to 922,340 YLLs each year.

### Economic burden

Cigarette smoking costs Nigeria ₦526.45 billion (approx. USD 1.7 billion) annually in direct treatment, which is equivalent to 0.36% of GDP and 9.63% of the country's annual healthcare budget. This burden is mainly attributable to COPD (63%), stroke events (12%), and cardiovascular diseases (6%). Additional indirect costs (productivity losses due to disability, premature death, and informal caregivers) total ₦107 billion. Informal caregivers are projected to represent ₦ 59 billion, while ₦ 24.3 and ₦ 23.8 billion are the consequence of disability and premature deaths, respectively, summing up, these costs represent 0.44% of the GDP.

In sum, the total economic burden account ₦ 634 billion considering direct treatment costs, productivity losses (due to early mortality and disability) and informal caregiving cost. In Nigeria, the tax revenue generated by the sale of cigarettes (and other tobacco products) was

**Table 3. Smoking-attributable deaths, events, and directs costs.**

| Tobacco-related conditions | Total deaths | Smoking- attributable deaths | | | Total events | Smoking- attributable events | | | Direct medical cost (in millions) | | Smoking- attributable costs | |
|---|---|---|---|---|---|---|---|---|---|---|---|---|
| | | N | % of Total deaths | % of Total smoking attributable deaths | | N | % of Total events | % of Total smoking attributable events | Total costs ₦ | Attributable costs ₦ | % of attributable cost from total | % of contribution of desease to total cost |
| **Cardiovascular diseases** | **72225** | **6616** | **9** | **23** | **95704** | **11150** | **12** | **8.75** | ₦ 242.413,19 | ₦ 33.248,71 | **14%** | **6%** |
| Ischemic Heart Disease | 49830 | 5067 | 10 | 17.5 | 95704 | 11150 | 12 | **8.75** | | | | |
| CV death of non-ischemic cause | 22395 | 1549 | 7 | 5.5 | NA | NA | NA | NA | | | | |
| **Stroke** | **44275** | **3767** | **9** | **13** | **100989** | **11477** | **11** | **9** | ₦ 432.209,20 | ₦ 61.109,98 | **14%** | **12%** |
| **Lung cancer** | **1255** | **843** | **67** | **3** | **1376** | **906** | **66** | **0.7** | ₦ 17.891,26 | ₦ 11.472,37 | **64%** | **2%** |
| **Pneumonia/ influenza** | **30442** | **3093** | **10** | **11** | **366013** | **31663** | **9** | **24.8** | ₦ 22.418,32 | ₦ 1.939,39 | **9%** | **0%** |
| **COPD** | **13162** | **8311** | **63** | **29** | **146411** | **68937** | **47** | **54** | ₦ 539.013,57 | ₦ 338.583,48 | **63%** | **63%** |
| **Other cancers** | **19202** | **2923** | **15** | **10** | **26872** | **3726** | **14** | **3** | ₦ 186.584,51 | ₦ 19.276,20 | 11% | 12% |
| Mouth and pharyngeal cáncer | 1954 | 890 | 46 | 3 | 2518 | 1134 | 45 | 1 | | | | |
| Esophageal cáncer | 624 | 269 | 43 | 1 | 735 | 320 | 44 | 0.2 | | | | |
| Stomach cáncer | 2 060 | 219 | 11 | 0.8 | 2401 | 250 | 10 | 0.2 | | | | |
| Pancreatic cáncer | 1947 | 246 | 13 | 0.9 | 2110 | 265 | 13 | 0.2 | | | | |
| Kidney cáncer | 481 | 56 | 12 | 0.2 | 575 | 67 | 12 | 0.1 | | | | |
| Laryngeal cáncer | 1002 | 635 | 63 | 2 | 1282 | 805 | 63 | 0.6 | | | | |
| Leukemia | 1634 | 128 | 8 | 0.4 | 2090 | 162 | 8 | 0.1 | | | | |
| Bladder cáncer | 683 | 151 | 22 | 0.5 | 943 | 202 | 21 | 0.2 | | | | |
| Cervical cáncer | 8817 | 329 | 4 | 1.1 | 14218 | 521 | 4 | 0.4 | | | | |
| **Secondhand smoking and other causes** | **3322** | **3322** | **100** | **11** | NA | NA | NA | NA | NC | ₦ 60.827,19 | | NA |
| **Total** | **183883** | **28876** | **16** | **100** | **737366** | **127859** | **17** | **100** | ₦ 1.440.530,05 | ₦ 526.457,32 | 36% | 100 |

AMI: acute myocardial infarction, ₦: Nigerian Naira, COPD: chronic obstructive pulmonary disease, CV: cardiovascular, NA: not applicable, U$D: US dollars. * Exchange rate per dollar U$D 1 = ₦306.92.

around ₦36 billion in 2019 [39], which covered only 6.9% of the direct medical costs of smoking, or 5.7% of the total financial burden.

## The impact of raising tobacco taxes

Table 5 shows that by increasing the price of cigarettes by 50%, we could prevent more than 30,000 deaths, 13,000 heart diseases, 5,562 new cancers, and 21,049 strokes over the next ten years. Furthermore, around ₦ 966,615 million in financial resources could be generated, a figure that is derived from savings in healthcare expenditures (₦ 474,712 million), productivity loss costs and informal caregiver costs avoided (₦ 63,688 million and ₦ 59,147, respectively), and increased fiscal revenue by tobacco tax collection (₦ 369,068 million). It is worth to clarify that these benefits would be possible explained by an increase of 168% on tobacco taxes,

**Table 4. Years of life lost (YLLs) due to premature mortality, disability, and total DALYs.**

| Disability-adjusted life-years (DALY) components | Women | Men | Total | % |
|---|---|---|---|---|
| Years of Life Lost due to premature mortality | 196618 | 431683 | 628302 | 77% |
| Years of life lost due to disability | 60661 | 127267 | 187929 | 23% |
| **Total DALY** | **257279** | **558951** | **816230** | **100%** |
| **YLLs due to premature mortality by disease group** | | | | |
| Cardiovascular disease | 39032 | 87590 | 126623 | **20.2%** |
| Stroke | 47156 | 59418 | 106574 | **17%** |
| Pneumonia /influenza | 22535 | 43944 | 66479 | **10.6%** |
| COPD | 48731 | 119418 | 168149 | **26.8%** |
| Lung cancer | 5947 | 21190 | 27137 | **4%** |
| Other cancers | 23041 | 74197 | 97237 | **15.4%** |
| **Total YLLs** | **196618** | **431683** | **628301** | **100.0%** |
| **Differential QALY per person in relation to a never-smoker** | | | | |
| **Smoking status** | **Women** | **Men** | | |
| Smoker | -5.83 | -5.49 | | |
| Ex-smoker | -1.93 | -2.45 | | |

COPD: chronic obstructive pulmonary disease, DALY: disability-adjusted life-years, QALY: Quality-adjusted Life Years, YLL: Years of Life Lost.

**Table 5. Economic consequences of smoking and the potential effects of price increase– 2020.**

| *Economic consequences of smoking* | |
|---|---|
| *Category* | *₦ (millions)* |
| Total health expenditure (THE) | 4,422,604 |
| Gross domestic product (GDP) | 121,167,234 |
| Tobacco-tax collection | 36,300 |
| Smoking-attributable direct costs of treatment | 526,457 |
| Treatment costs as % of GDP | 0.36% |
| Treatment costs as % of THE | 9.63% |
| % of treatment costs recovered with taxes | 6.90% |
| % of total costs recovered with tax | 5.73% |

| Scenarios for price increase: 10 years effect for different % increase | | | |
|---|---|---|---|
| *% increase in final price of a package* | *25%* | *50%* | *75%* |
| Deaths prevented | 15 454 | 30 908 | 46 361 |
| Heart disease avoided | 6 392 | 12 784 | 19 175 |
| Number of Strokes avoided | 10 525 | 21 049 | 31 574 |
| New cases of cancer avoided | 2 781 | 5 562 | 8 342 |
| New cases of COPD avoided | 23 919 | 47 838 | 71 757 |
| DALYs avoided | 520 374 | 1040 747 | 1561 121 |
| Health costs avoided | ₦237,356.00 | ₦474,712.00 | ₦712,068.00 |
| Informal caregivers costs avoided | ₦29,573.00 | ₦59,147.00 | ₦88,720.00 |
| Productivity losses avoided | ₦31,848.00 | ₦63,688.00 | ₦95,522.00 |
| Increase in tax collection | ₦222,385.00 | ₦369,068.00 | ₦440,050.00 |
| *Total economic benefit (in millions)* | *₦521,161.00* | *₦966,615.00* | *₦1,336,359.00* |

₦: Nigerian Naira, exchange rate ₦ 306 = U$D 1, DALY: disability-adjusted life-years, GDP: gross domestic product, THE: total health expenditure.

assuming a complete pass-through between price and excises. In addition, in a scenario of the potential increase of the illicit trade of tobacco products, there might remain 92% of the total economic gains after the price increase through taxes.

Two additional scenarios are presented, one as a conservative after a raise in prices of 25%, and another promising scenario where the increase of tobacco price is 75%. Regarding the former, the economic benefit could reach ₦ 521 billion with ₦ 222 billion being due to increase in the tax collection, reaching more than a half of the benefit but with an increase of the tax rate 84 pp. lower according to the current percentage of price that are. In the latter, achieving a 75% price increase would lead to an increase in tax collection of 120%, showing that there is still place to increase fiscal and health benefits at the same time, due to the low starting tax levels.

## Discussion

The results of this study show that Nigeria suffers from both a significant burden of disease and an economic burden associated with smoking. According to our findings, near 29,000 deaths and 800,000 DALYs are attributable to smoking in the country annually. Those deaths represent around 5% of all country deaths in one year.

These findings are in line with those reported by the Global Burden of Disease (2019) [1]. Although both (total number of deaths and DALYs estimates) are higher than the central values reported by this study [1], they do not exceed the upper values of the range reported (approx. 30,000 deaths and more than 850,000 DALYs).

On the other hand, the total economic burden was estimated at ₦ 634 billion, which represents almost half of a percentage point of the Nigerian GDP, with the cost of treating tobacco-related diseases counting for the 83% of that burden. Our results show that important benefits could be obtained from raising tobacco taxes. An increase of 50% of cigarette price through taxes could prevent more than 30,000 deaths as well as generate a total economic benefit of ₦ 966,614 million at ten years due to avoided treatment costs (50%), gains in tax revenue (38%), and averted indirect costs (12%).

Compared to other regions, Africa has paid little attention to tobacco use consequences and tobacco control policies. A possible explanation is the perceived low prevalence of smoking in Africa [5], as well as the urgent need to fight infectious diseases. For instance, Goodchild et al. [57] has estimated the global economic burden of diseases related to smoking using estimated data from a literature review, finding that 1.7% of deaths worldwide correspond to the African continent. Furthermore, the study reports that the total costs, direct and indirect costs as well, of smoking represented US $ 1,436 billion, being 1.8% of the global GDP, while Africa has direct health costs of US $ 15 billion (1% of their GDP). These differences among regions might be explained by the relatively lower prevalence of tobacco consumption.

For Nigeria, this study shows that the economic burden would rise to 0.45% of their GDP, which, as could be expected, is less burden than estimated in Goodchild et al. [57], due to their estimation on direct costs that rely on primarily high-income countries cost.

Another research studied the economic cost of smoking for South Africa, which amounted to 0.97% of the South African GDP in 2016, while the healthcare cost of smoking-related diseases was 4.1% of total South African health expenditure [58]. In Uganda [59] through a COI approach, the direct and indirect costs of tobacco were estimated to be USD 126.48 million, which is equivalent to 0.5% of GDP, a result similar to that of this study.

Previous research addressed some dimensions of the economic burden of tobacco consumption for Nigeria. Owoeye et al. 2015 estimated the total economic cost faced by patients, out-of-pocket, in Ibadan Hospitals using the prevalence-based method of the cost of illness

(COI) approach for four tobacco-related diseases, namely Stroke or Transient Ischemic attack, lung cancer, COPD, and tuberculosis base on a questionnaire made to 320 patients. The authors found that the mean cost of treating diseases related to smoked tobacco was ₦ 65,587, and using a prevalence-based analysis they concluded that the economic cost for patients of Nigeria would be ₦ 1,821,743 [60]. It should be clarified that these results are not strictly comparable with those presented in this research since the present work evaluates the total economic burden of disease for Nigeria.

Our study estimated that the informal caregivers suffer an economic cost of ₦ 59,174 million annually, representing 55% of the total indirect cost attributable to smoking and 9% of the total economic burden. Consequently, this result is consistent with other studies showing the importance of informal caregivers' health and economic burden in Nigeria. These studies show that 41% of informal caregivers experience a financial burden besides physical, psychological, and social burden [61, 62]. Additionally, according to the literature, most informal caregivers are in their young and active economic age, and they are predominantly females, who are wives and/or daughters [63], which could imply potential inequalities to the detriment of women due to the greater burden of care.

In 2017, Nigeria introduced a new scheme on tobacco taxation policy.

A special component of ₦20 per pack is included in this scheme, adding to the previous ad-valorem rate of 20% over the unit cost of production for the first year, and with further increases in 2018 and 2019, the price should reach ₦58 per pack of 20 cigarettes in 2020.

The amount of tax per package was doubled, but the tax percentage was still around 20% (including VAT), considerably lower than the WHO recommendation to be closer to 75% [64, 65]. Additionally, it is necessary to complement tax policies with other additional policies for tobacco control, such as those proposed by MPOWER, an initiative in which Nigeria is behind in the implementation of complementary strategies to control the tobacco epidemic [66].

The application of our model entails significant advantages that make it useful for decision-making in public health in Nigeria and broader Africa. First, its suitability for a context of scarcity of epidemiology and economic data like Nigeria's. Second, its ability to interrogate different dimensions of the tax burden (gender, age group, level of taxation) and evaluate the effectiveness of other policies like smoke-free air legislation, packaging, and advertising, not shown in this manuscript. Of note, although our study measures the disease burden of smoking-related diseases, it also considers their indirect costs by premature death, disability, and costs of informal care. Last, local information on costs and resource usage from hospitals of three different geographical regions in Nigeria was used for the modeling.

The study offers suggestions on how the government can raise tobacco taxes. Thus, the fiscal revenue would increase by 101%. Furthermore, our study suggests that 92% of the total economic benefit endure despite potential illicit trade increase. This result shed light on the tobacco industry's argument, which advocates against tobacco tax, arguing the potential increase in illicit trade, which often is overestimated by the industry [61]. Our results show that even in a pessimistic scenario of illicit trade, Nigeria will benefit from increasing tobacco taxes.

Because the same methodology was used by Pichon-Riviere et al. [13], one can make some comparisons between the tobacco burden in Latin America (LA) and Nigeria. As a percentage of GDP, Nigeria's direct costs for smoking conditions are 60% lower than the average for LA countries. However, the results obtained for a country such as Honduras, which is comparable to Nigeria in terms of GDP per capita, are similar. Nevertheless, the highest difference is related to the percentage of the direct medical costs recovered by fiscal revenues.

On average, the LA economy recovers 36% of its direct medical costs through taxes, while Bolivia only collects only 6%, similar to our estimates for Nigeria. This situation highlights the necessity to strengthen tobacco tax policies in Nigeria.

Based on Nigeria's 2017 Voluntary National Review (VNR), which illustrates the development priorities of the President's office over Sustainable Development Goals (SDG) [62], this study provides relevant evidence for serving all objectives within the study area.

SDG-3 calls for reducing non-communicable diseases premature mortality by one-third, which can only be achieved with tobacco control policies, through prevention, treatment, and promoting mental health and well-being, and strengthening the prevention and treatment of substance abuse, among others.

Additionally, we estimated the benefits associated with informal care costs avoided (which tend to be unpaid activities frequently led by women) useful to address the SDG-5 (that aims to eliminate all forms of discrimination and violence against women). A South-South collaboration process between IECS (Latin America) and CSEA (Africa) was used to identify SDG-17 (that refers to the need for cross-country collaboration).

As strengths of our work, we could affirm that this is the first study to show the burden of disease -where deaths, disease events, and utility values are taking into consideration- as well the corresponding economic burden (direct medical costs, and indirect costs including productivity loss costs and caregivers cost) attributable to tobacco consumption in Nigeria. In addition, our study estimated the impact of different scenarios of tobacco price increases through taxes, including an additional scenario including the potential effects of the illicit trade in the country. This complete panorama about the burden of tobacco consumption and the benefits of the tobacco tax increase should help policymakers act.

Some limitations of our study should be mentioned. First, we used smoking prevalence data from the GATS 2012 survey; the country did not update this representative survey. With more actual smoking prevalence data, the results might differ from those reported in the present study. Second, Nigeria is a diverse country, and data may vary in its different geographical regions. However, so far, we do not count on enough information detail to undertake subnational estimations.

Third, due to the lack of local/regional data regarding risk relative values for the quantitative association between each smoking condition (smokers, former smokers, and passive smoking) with each tobacco-associated disease, we decided to use data from well-designed study cohorts carried out in the U.S. We acknowledge that the extrapolation of the U.S. estimates of, for example, the consequences of passive smoking, may be different from the reality in Nigeria given, mainly, the wide differences in population characteristics. Fourth, no country-representative sampling was feasible. However, the large hospitals surveyed covered three of the main geopolitical and cultural subregions in the country.

Five, our estimation of the economic burden does not include the potential non-medical costs of treatment as transportation, childcare, per diem that mostly run at the expense of the patient. Finally, our model does not consider the socioeconomic equity dimensions (e.g., by income quintiles or gender, out-of-pocket expenditure), so it was not feasible to estimate which specific subpopulations would benefit more from increases in tobacco taxes. This remains a gap for future research.

In conclusion, our findings show that a tobacco tax increase could translate into health benefits and reduction in direct and indirect costs attributable to tobacco.

## Supporting information

**S1 Appendix. Relative risks of mortality for smokers and ex-smokers for each tobacco-related condition, by sex (in reference to never-smokers).**
(XLSX)

**S2 Appendix. Utility values by disease.**
(XLSX)

**S3 Appendix. Smoking prevalence and total population by single age and sex.**
(XLSX)

## Author Contributions

**Conceptualization:** Ariel Bardach, Andrés Pichon-Riviere, Alfredo Palacios.

**Data curation:** Agustín Casarini, Federico Rodriguez Cairoli, Adedeji Adeniran, Marco Castradori, Precious Akanonu.

**Formal analysis:** Agustín Casarini, Federico Rodriguez Cairoli, Andrés Pichon-Riviere, Alfredo Palacios.

**Funding acquisition:** Ariel Bardach, Adedeji Adeniran, Andrés Pichon-Riviere, Alfredo Palacios.

**Investigation:** Ariel Bardach, Federico Rodriguez Cairoli, Adedeji Adeniran, Precious Akanonu, Chukwuka Onyekwena, Andrés Pichon-Riviere, Alfredo Palacios.

**Methodology:** Ariel Bardach, Agustín Casarini, Federico Rodriguez Cairoli, Adedeji Adeniran, Marco Castradori, Precious Akanonu, Chukwuka Onyekwena, Natalia Espinola, Andrés Pichon-Riviere, Alfredo Palacios.

**Project administration:** Ariel Bardach, Andrés Pichon-Riviere, Alfredo Palacios.

**Resources:** Precious Akanonu, Andrés Pichon-Riviere, Alfredo Palacios.

**Software:** Andrés Pichon-Riviere.

**Supervision:** Ariel Bardach, Adedeji Adeniran, Natalia Espinola, Andrés Pichon-Riviere, Alfredo Palacios.

**Validation:** Ariel Bardach, Adedeji Adeniran, Precious Akanonu, Chukwuka Onyekwena, Natalia Espinola, Andrés Pichon-Riviere, Alfredo Palacios.

**Visualization:** Ariel Bardach, Agustín Casarini, Federico Rodriguez Cairoli, Adedeji Adeniran, Marco Castradori, Precious Akanonu, Chukwuka Onyekwena, Natalia Espinola, Andrés Pichon-Riviere, Alfredo Palacios.

**Writing – original draft:** Ariel Bardach, Agustín Casarini, Federico Rodriguez Cairoli, Adedeji Adeniran, Marco Castradori, Precious Akanonu, Chukwuka Onyekwena, Natalia Espinola, Andrés Pichon-Riviere, Alfredo Palacios.

**Writing – review & editing:** Ariel Bardach, Agustín Casarini, Federico Rodriguez Cairoli, Adedeji Adeniran, Marco Castradori, Precious Akanonu, Chukwuka Onyekwena, Natalia Espinola, Andrés Pichon-Riviere, Alfredo Palacios.

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
