## [Decision Letter · Decision Letter 0]

16 Sep 2021

PONE-D-21-15723The estimated benefits of increasing cigarette prices through taxation on the burden of disease and economic burden of smoking in Nigeria: A modeling studyPLOS ONE

Dear Dr. Rodriguez Cairoli,

Thank you for submitting your manuscript to PLOS ONE. After careful consideration, we feel that it has merit but does not fully meet PLOS ONE’s publication criteria as it currently stands. Therefore, we invite you to submit a revised version of the manuscript that addresses the points raised during the review process.

We look forward to receiving your revised manuscript.

Kind regards,

Rashidul Alam Mahumud, MPH, MSc, PhD

Academic Editor

PLOS ONE

Journal Requirements:

2. Thank you for submitting the above manuscript to PLOS ONE. During our internal evaluation of the manuscript, we found significant text overlap between your submission and the following previously published works, some of which you are an author.

- http://cseaafrica.org/wp-content/uploads/2021/03/Health-Burden-Economic-Cost-of-smoking-In-nigeria-2.pdf

 The text that needs to be addressed involves the Results and Discussion sections.

Please revise the manuscript to rephrase the duplicated text, cite your sources, and provide details as to how the current manuscript advances on previous work. Please note that further consideration is dependent on the submission of a manuscript that addresses these concerns about the overlap in text with published work.

Additional Editor Comments (if provided):

Please revise your manuscript according to the reviewer's feedback and suggestions.

Reviewers' comments:

Reviewer's Responses to Questions

**Comments to the Author**

1. Is the manuscript technically sound, and do the data support the conclusions?

Reviewer #1: Yes

Reviewer #2: Partly

2. Has the statistical analysis been performed appropriately and rigorously? 

Reviewer #1: Yes

Reviewer #2: Yes

3. Have the authors made all data underlying the findings in their manuscript fully available?

Reviewer #1: Yes

Reviewer #2: Yes

4. Is the manuscript presented in an intelligible fashion and written in standard English?

Reviewer #1: Yes

Reviewer #2: Yes

5. Review Comments to the Author

Reviewer #1: The paper is nicely done and very informative. It fills a huge gap in the tobacco control research literature. I have provided detailed comments in the review report to help improve the manuscript to a publishable form.

Thank you for the excellent work.

Reviewer #2: The main comment for this paper concerns the weighting procedure for the costs. To make them nationally representative, the weighting procedure briefly mentioned in line 190 seems to suggest that weighting up the costs based on population size in each region is equivalent to weighting it to make it nationally representative. This is may necessarily be the case. Short of more detail, it appears the population weights used would make the costs regionally representative. To make them nationally representative would require more detail on the sampling procedure used for the selection of the four hospitals selected for the primary data collection. As this is not described in the paper, it is difficult to assess whether weighting to be nationally representative is possible. If not, I would suggest including it at least as a limitation.

6. PLOS authors have the option to publish the peer review history of their article (what does this mean?). If published, this will include your full peer review and any attached files.

Reviewer #1: **Yes: **Nigar Nargis

Reviewer #2: No

---

## [Author Response · Author response to Decision Letter 0]

27 Nov 2021

Main comments

Reviewer #1: The paper is nicely done and very informative. It fills a huge gap in the tobacco control research literature. I have provided detailed comments in the review report to help improve the manuscript to a publishable form.Thank you for the excellent work.

Response. Thank you for your appreciation and these contributions.

Reviewer #2:The main comment for this paper concerns the weighting procedure for the costs. To make them nationally representative, the weighting procedure briefly mentioned in line 190 seems to suggest that weighting up the costs based on population size in each region is equivalent to weighting it to make it nationally representative. This is may necessarily be the case. Short of more detail, it appears the population weights used would make the costs regionally representative. To make them nationally representative would require more detail on the sampling procedure used for the selection of the four hospitals selected for the primary data collection. As this is not described in the paper, it is difficult to assess whether weighting to be nationally representative is possible. If not, I would suggest including it at least as a limitation.

Response. Thank you. These four hospitals surveyed were selected with the sole purpose of covering three distinct geopolitical and cultural zones across the country. Specifically, the selected places covered the North-Central Region (covered by the hospitals in Abuja), the South-western region (Ibadan), and the Southeast (Enugu). Based on access to treatment facilities, these institutions are the main hospitals of their respective regions. Although no representative sampling was used, this fact helps to account for the social and economic disparities in different subregions. We added the following explanation in the discussion section, underlining this limitation:

“Fourth, no country-representative sampling was feasible. However, the large hospitals surveyed covered three of the main geopolitical and cultural subregions in the country”

Introduction

1. Page 3 lines 44-45: The Introduction starts with the citation of the tobacco-attributable deaths and disabilities for 2017. More recent estimates of tobacco-attributable deaths and disabilities are available for 2019. Please see the quote below: 

“Globally in 2019, smoking tobacco use accounted for 7·69 million (7·16–8·20) deaths and 200 million (185–214) disability-adjusted life-years, and was the leading risk factor for death among males (20·2% [19·3–21·1] of male deaths). 6·68 million [86·9%] of 7·69 million deaths attributable to smoking tobacco use were among current smokers.”

Reference: GBD 2019 Tobacco Collaborators. Spatial, temporal, and demographic patterns in prevalence of smoking tobacco use and attributable disease burden in 204 countries and territories, 1990–2019: a systematic analysis from the Global Burden of Disease Study 2019. Lancet 2021; 397: 2337–60 Published Online May 27, 2021 https://doi.org/10.1016/ S0140-6736(21)01169-7.

Response. Thank you for this contribution. Now we are referencing this latest study. 

2. Page 4 line 70: In the cost-of-illness approach to the estimation of the economic burden of tobacco use, the direct medical costs include the treatment costs incurred by both the public health system and the out-of-pocket expenses of the patients and their families. By accounting the costs of “treatment incurred on the health system” only, this paper underestimates the direct medical costs. Please clarify how the out-of-pocket expenses that generally account for a major fraction of total medical expenditures were considered.

Response. Thanks for the comment. We estimated the cost of health events regardless of who is covering them (health system or patient through OOP). We agree that we are focusing on the direct medical cost, not considering potential out-of-pocket expenses (transportation expenses, childcare, per diem, etc.), as described in methods. This is a limitation of the study and is now adequately acknowledged. However, we are reporting the opportunity costs of informal caregiver and productivity losses, so, importantly, this is the first study to report these indirect costs in Nigeria

Methods

1. Page 4 lines 92-94: “The risk of acute and chronic events is estimated from 93 the baseline risk in non-smokers multiplied by the age, gender, and condition-specific relative 94 risks (RR) for smokers and ex-smokers.” Based on the citation (14), it seems that the authors used the RRs from a study in the U.S. Please explain why the U.S. based RRs were used and how these estimates were validated for Nigeria. I would recommend the authors use evidence available from countries that are comparable to Nigeria or at least from the same region. They can justify the use of U.S. estimates if none of these estimates are available and if the previous studies based on the same model, such as citations (8-13), used the same parameters.

Response. Thank you for this contribution. This analysis uses the same relative risk (RR) parameters as previous studies (citations: 8-13). This is mainly because the primary source of these parameters (14), is, as far as we know, the most well-conducted cohort study with enough information carried out in the field. No African cohort study assesses and reports this information (each RR for each disease and smoking condition). Thus, now we include a sentence in the discussion section to explain this limitation.

2. Page 5 lines 102-105: Please use a reference and the ICD-10 codes for the list of health conditions considered for the study. 

Response. Thank you for this contribution. We have added it now. 

3. Page 5 lines 106-110: The extrapolation of the U.S. estimates of the consequences of passive smoking to Nigeria may be far off the realities in Nigeria given the wide differences in population characteristics, disease events, and healthcare systems in the two countries. Similar to Comment 1 on Methods, I would recommend the authors use evidence available from countries that are comparable to Nigeria or at least from the same region. They can justify the use of U.S. estimates if none of these estimates are available and if the previous studies based on the same model, such as citations (8-13), used the same parameters.

Response. Thank you for this contribution. As was explained before, this analysis uses the same parameters as previous studies (citations: 8-13). To our knowledge, there is no African cohort study that reports a second-hand smoking value parameter to be used directly by our model. We inserted a sentence in the discussion section to explain this limitation. 

We had identified the recent Yousuf et al study which estimated mortality attributable to secondhand tobacco exposure in several regions, including sub-Saharan Africa. They used a secondhand smoke index (SHSI). SHSI measures the number of smokers during an average period of 24 years for each non-smoker who died. For Sub-Saharan Africa, this value was approximately 57 active smokers for a 24-year period. Applying this number to the annual number of smokers as estimated by GATS 2012, and then dividing it by 24 years, gives an approximate total of 2,500 deaths per year. This result is within +/- 20% of our estimated results for deaths due to SHS.

4. Page 6 lines 129-130: It is not clear how the variation in smoking prevalence (page 5 line 121) was translated into variation in consumption to estimate the effect on tax revenue.

Response. Thanks. The variation in the consumption was obtained multiplying the price elasticity by the assumed change in the price, and this was added to the manuscript. The impact on tax revenue is equal to change in consumption times the price change weighted by the proportion of the price that corresponds to taxes. We clarified this in the manuscript, as well as the formula for the computation of the change in tax revenue

We first estimate the variation in consumption (based on elasticity) and then translate that into a variation in prevalence according to the Ip parameter (more information about this parameter and the estimation of benefits in Pichon-Riviere, A. et al. (2020) ‘The health and economic burden of smoking in 12 Latin American countries and the potential effect of increasing tobacco taxes: an economic modeling study’, The Lancet. Global health, 8(10), pp. e1282–e1294.) using the following formula:

Prevalence=PrevB+(Ed*∆P*Iρ*PrevB)

5. In page 7 lines 155, 158, 159 and un-Table 2, the notation of the local currency is different (₦ or NGN). The Results section in the Abstract, on the other hand, mentions Naira. Please make it consistent across the manuscript. When it is used the first time in the manuscript, it should be mentioned that it is Nigerian local currency Naira. 

Response. Thanks, we adopt the sign ₦ for referring Nigerian Naira’s.

6. Page 10 lines 207-210: The household expenditure needs to be converted to per capita household expenditure by dividing total expenditure by household size to be assigned to individuals’ annual income. It is not clear whether this conversion was done or not.

Response. Thanks. The household expenditure was divided the total amount of workers (members that were working for someone out of the family or own entrepreneur) within the household. Now this is explained in line 232.

7. Page 10 lines 210-213: IMF provide projection of per capita GDP for 4/5 years from current year when the World Economic Outlook data is updated each year. These projections are based on models that consider several growth factors and their recent trends. I would recommend authors use the growth rate projected in these data instead of using mean of annual growth rates from 1960 to 2019 which may be biased due to the long horizon of 60 years when many low- and middle-income countries have made major strides in economic growth.

Response. Thank you. What you raise is also an interesting possibility, albeit not free from biases, since it aims to estimate a short-term scenario. Biases could arise from the economic cycle and external shocks, as is the COVID outbreak case, which cannot be captured well with this kind of forecast. We consider an extensive series of annual GDP per capita growth in constant dollars because we understand could be more representative of the reality of the country and could be more accurate for estimating the growth of income for the lifetime of our cohort of individuals due to be less susceptible to macroeconomic cycles (after the first year of covid, forecasts models would predict high-income growth) and would represent better the long-term trend of GDP . We also decided to follow the Pinto 2019 study methodology for Brazil.(Pinto, M. et al. (2019) ‘Burden of smoking in Brazil and potential benefit of increasing taxes on cigarettes for the economy and for reducing morbidity and mortality’, Cadernos de saude publica, 35(8), p. e00129118.)

8. Page 10 line 220: What does the term “profit associated with the disease” imply? Please clarify.

Response. Thank you for this contribution. It was a translation error, it refers to utility, not “profit”. We edited the manuscript accordingly.

Results

1. How do the deaths and events attributable to tobacco use estimated in this study compare the estimates available in the Global Burden of Disease Study? These can be compared to show the validity of the study findings.

Response. Thank you for this contribution. Now we have added a sentence in the discussion section:

“These findings are in line with those reported by the Global Burden of Disease (2019). Although both (total number of deaths and DALYs estimates) are higher than the central value reported by this study, they do not exceed the upper value of the range reported (approx. 30,000 deaths and more than 850,000 DALYs)”

2. In Table 3, the last two columns present % of attributable cost whereas the head of the column state “Smoking-attributable costs (millions)”. Please consider revising the head.

Response. Thank you for this contribution. We changed it according to this suggestion.

3. Table 3 would be more legible in landscape page orientation and broken into two pages.

Response. Thank you for this contribution. We changed it according to this suggestion.

 4. The two columns reporting Economic burden (in millions) in Table 3 are not discussed in the text. The title of the table says, “direct costs for the healthcare system”. So, I am assuming these columns refer to the medical costs. But the economic burden implies indirect costs or the productivity losses as well. Are the indirect costs included in these estimates? If yes, then the title needs to be revised accordingly.

Response. The title of the column was changed accordingly, also some of the figures have changed.

5. Page 13 lines 256-258: The estimation of the burden of passive smoking is explained in the Methods section. But the burden of perinatal disease and accidents related to smoking is mentioned for the first time in the paper in the Results section. How were these estimates obtained?

Response. Thank you for this contribution. Now we have added information regarding this point in the methods section:

“Although the model does not assess the consequences of passive smoking and the main smoking-related perinatal causes (low weight or low size at birth, respiratory distress syndrome, and sudden infant death syndrome) directly, the potential years of life lost, deaths, and costs associated with it were incorporated using estimates reported in US studies (15).”

6. Page 14 lines 264-276: The costs estimate reported in the section on Economic Burden do not match the economic burden estimates presented in Table 3. It seems these estimates are presented in Table 5. Please ensure the correct and consistent reporting of the cost estimates. 

Response. Thank you for this contribution. Estimates in table 5 were corrected in the manuscript.

7. The estimates in Table 5 refer to 10-year effects. It is not clearly mentioned in the Methods section that the simulation was done for the 10-year period following the tax increase.

Response. Thank you for this contribution. Clarification was added to methodology section

8. Table 5 reports the effects of price increases by 25%, 50% and 75% and discusses the results of only 50% price increases. It is not clear why the authors wanted to include the estimates of the effects of 25% and 75% price increases and hence those estimates seem redundant. If they are keen on keeping these estimates, I would recommend making the point that higher price increases would lead to greater cost saving and revenue gain.

Response. Thank you for this contribution. An explanation on the scenarios considered was added.

Discussion

9. Page 16 lines 306-312: The authors refer to the global study Goodchild et al to present the global estimates. How do the cost estimates for Nigeria in this global study (presented in the Supplementary Material) compared to the estimates obtained in the present study?

Response. Thank you for this contribution. In the Goodchild study there were no estimations for Nigeria. For that reason we decided to discuss our results compared to the regional scenario. Additionally, the estimation of cost in Goodchild was based in predicted costs exploiting the correlation between smoking attributable deaths and expenditure mostly in high income countries. So, there could be the reason why the estimated burden for Africa it is greater than the one found for Nigeria in our study. This explanation was added to the manuscript.

1. Page 18 lines 354-357: This paragraph reporting amount of tax increase by 168% leading to 50% price increase and 101% increase in fiscal revenue belongs to the Results section.

Response. Thank you for this contribution. This comment was added to result section.

2. Page 18 lines 364-366: Is the comparison of direct cost in Nigeria to the average of LA countries in absolute levels? What is the unit of this measurement? For cross-country comparison, the estimates should be converted to per capita terms and PPP dollars to adjust for population size and differences in purchasing power of dollars.

Response. Thank you for this contribution. The comparisons are based on relative figures (direct cost as % of GDP, %medical cost covered by tax collection) so there’s no need of converting the terms on PPP dollars or adjusting population size.

3. Page 10 lines 400-402: This paper does not estimate the net economic benefits. Because there is a cost of implementation of tobacco tax and the net benefit should exclude this cost that was not taken into consideration. I suggest dropping the part “resulting in substantial net economic benefits for Nigeria” from the concluding statement.

Response. Thank you for this contribution. The word “net” was taken out of the manuscript.

Supporting Information 

The data sources for the tables in the Appendices S1, S2 and S3 should reported with full citation

Response. Done. Thank you for this contribution.

---

## [Decision Letter · Decision Letter 1]

27 Jan 2022

PONE-D-21-15723R1The estimated benefits of increasing cigarette prices through taxation on the burden of disease and economic burden of smoking in Nigeria: A modeling studyPLOS ONE

Dear Dr. Rodriguez Cairoli,

Thank you for submitting your manuscript to PLOS ONE. After careful consideration, we feel that it has merit but does not fully meet PLOS ONE’s publication criteria as it currently stands. Therefore, we invite you to submit a revised version of the manuscript that addresses the points raised during the review process.

We look forward to receiving your revised manuscript.

Kind regards,

Rashidul Alam Mahumud, MPH, MSc, PhD

Academic Editor

PLOS ONE

Journal Requirements:

Reviewers' comments:

Reviewer's Responses to Questions

**Comments to the Author**

1. If the authors have adequately addressed your comments raised in a previous round of review and you feel that this manuscript is now acceptable for publication, you may indicate that here to bypass the “Comments to the Author” section, enter your conflict of interest statement in the “Confidential to Editor” section, and submit your "Accept" recommendation.

Reviewer #2: All comments have been addressed

2. Is the manuscript technically sound, and do the data support the conclusions?

Reviewer #2: Yes

3. Has the statistical analysis been performed appropriately and rigorously? 

Reviewer #2: Yes

4. Have the authors made all data underlying the findings in their manuscript fully available?

Reviewer #2: Yes

5. Is the manuscript presented in an intelligible fashion and written in standard English?

Reviewer #2: Yes

6. Review Comments to the Author

Reviewer #2: Response adequately addresses my main comment. The authors include a line in the limitations to highlight the point.

7. PLOS authors have the option to publish the peer review history of their article (what does this mean?). If published, this will include your full peer review and any attached files.

Reviewer #2: **Yes: **Zunda Chisha

---

## [Author Response · Author response to Decision Letter 1]

12 Feb 2022

Dear Editor

We apologize for our error regarding the references section. We have now submitted an untracked version with revised citations.

If there is anything else missing please let us know.

Best regards

---

## [Editor Report · Decision Letter 2]

17 Feb 2022

The estimated benefits of increasing cigarette prices through taxation on the burden of disease and economic burden of smoking in Nigeria: A modeling study

PONE-D-21-15723R2

Dear Dr. Rodriguez Cairoli,

We’re pleased to inform you that your manuscript has been judged scientifically suitable for publication and will be formally accepted for publication once it meets all outstanding technical requirements.

Kind regards,

Rashidul Alam Mahumud, MPH, MSc, PhD

Academic Editor

PLOS ONE
---

## [Editor Report · Acceptance letter]

22 Feb 2022

PONE-D-21-15723R2 

The estimated benefits of increasing cigarette prices through taxation on the burden of disease and economic burden of smoking in Nigeria: A modeling study 

Dear Dr. Rodriguez Cairoli:

I'm pleased to inform you that your manuscript has been deemed suitable for publication in PLOS ONE. Congratulations! Your manuscript is now with our production department. 

Kind regards, 

on behalf of

Dr. Rashidul Alam Mahumud 

Academic Editor

PLOS ONE